# Learning to Handle Large Obstructions in Video Frame Interpolation

## ABSTRACT

Video frame interpolation based on optical flow has made great progress in recent years. Most of the previous studies have focused on improving the quality of clean videos. However, many real-world videos contain large obstructions which cause blur and artifacts making the video discontinuous. To address this challenge, we propose our Obstruction Robustness Framework (ORF) that enhances the robustness of existing VFI networks in the face of large obstructions. The ORF contains two components: (1) A feature repair module that first captures ambiguous pixels in the synthetic frame by a region similarity map, then repairs them with a cross-overlap attention module. (2) A data augmentation strategy that enables the network to handle dynamic obstructions without extra data. To the best of our knowledge, this is the first work that explicitly addresses the error caused by large obstructions in video frame interpolation. By using previous state-of-the-art methods as backbones, our method not only improves the results in original benchmarks but also significantly enhances the interpolation quality for videos with obstructions.

## CCS CONCEPTS

• **Computing methodologies → Reconstruction**.

## KEYWORDS

Video Frame Interpolation, Obstruction Handling, Cross-attention

## 1 INTRODUCTION

Video frame interpolation (VFI) synthesizes intermediate images from given image sequences. Related algorithms are widely implemented in real-world applications, such as video super-resolution [13, 38], slow motion generation [1, 10, 26], novel view synthesis [11, 43] and video compression [21].

Deep learning-based methods have achieved remarkable results on VFI benchmarks. Previous methods utilize kernels [2, 5, 20] or optical flow [9, 15, 22, 25, 28, 29, 41] to align each object with its location in the neighboring frames.

Progress so far mainly focuses on interpolation quality [22, 28] and efficiency [7, 15]. Most existing video frame interpolation methods assume that scene objects are in clear view without any external interference. However, in many real-world applications, complex obstructions are often present due to various factors. For example,

*ACM MM, 2024, Melbourne, Australia*

© 2024 Copyright held by the owner/author(s). Publication rights licensed to ACM.
ACM ISBN 978-x-xxxx-xxxx-x/YY/MM
https://doi.org/10.1145/nnnnnnn.nnnnnnn

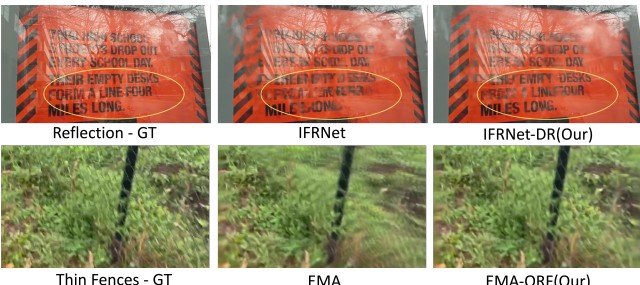

**Figure 1: Predictions in real-world obstruction scenes, standard approaches (*i.e.*, IFRNet [15], EMA [41]) fail due to occlusions and motion error, our method makes the models robust in those conditions.**

| Reflection - GT | IFRNet | IFRNet-DR(Our) |
| Thin Fences - GT | EMA | EMA-ORF(Our) |

video captured when driving, during sports and in outdoor settings may involve obstructions caused by fences, raindrops or reflections. Obstructions can cause large blurred areas and artifacts in the interpolated frames degrading the visual quality and temporal consistency of the output [1, 25]. As a result, applying previous VFI methods directly on videos with obstructions leads to subpar performance. Calculating the motion field in a video with obstructions is difficult, as pixels of an object often lack matches between two input frames.

Previous methods [3, 9, 15, 30, 42] directly feed the input frames into the network and hope the network can fix the occlusions as shown in Figure 2 (a). Some recent methods apply attention-based mechanisms [22], which could globally match features to better handle occlusions as depicted in Figure 2 (b). However, those methods still generate blur and artifacts in scenes with large occlusions caused by obstructions (see examples in Figure 1). This shows that mere awareness of occlusions is insufficient for reconstructing large occlusions caused by obstructions.

A more explicit approach is needed to identify blurred regions within the features and subsequently recover them. To achieve this goal, we propose an Obstruction Robustness Framework to handle videos with obstructions. First, we observe that a majority of errors caused by large obstructions are dense and localized around the respective obstruction. Intuitively, this issue can be addressed by identifying the error-prone region and selectively rectifying only this specific area. Thus, we propose a feature repair module which identifies the potentially ambiguous pixels or regions by a Region Similarity Map (RSM). The RSM assesses pixel similarity across warped adjacent input frames to indicate dissimilar regions that are prone to errors. We specifically repair these dissimilar regions utilizing a cross-overlap attention (COA) module. The key insight of the COA is that we can directly restore the erroneous regions within the synthetic frame by examining analogous areas within the

Figure 2: Comparison between current VFI methods and our approach. (a) the basic VFI model structure (*i.e.,* IFRNet [15], and ABME [30]). (b) the VFI model with global attention for long-range feature matching(*i.e.,* VFIformer [22]). (c) our explicit feature repaired framework first identifies the error region, and then fix it by feature repair.

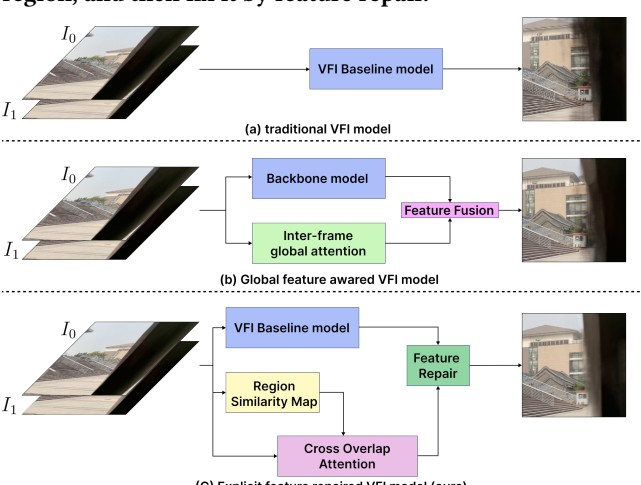

input image. The COA effectively matches and aggregates semantic information of objects from the input frames into the synthesized frame. We then employ these semantic features to correct the ambiguous regions in the synthesized frame, as previously delineated by the RSM (see in Figure 2 (c)).

Second, we introduce a data augmentation strategy Dynamic Mask Distractions (DMD), which captures dynamic obstructions from real scenes and integrates them into clean videos, thereby generating extra training pairs. DMD differs from prior data augmentation strategies in video frame interpolation, which typically involves the artificial insertion of shapes [17] or low-level modifications [15, 22, 35]. Instead, DMD utilizes semantically meaningful obstructions derived from real-world scenarios. Moreover, we animate these obstructions to mimic the movement in natural settings. We demonstrate DMD can improve the robustness of existing methods on both videos with and without obstructions without any new videos.

To prove the efficiency of our method, we apply it to various state-of-the-art VFI models. We then construct a special test set called Real World Obstructions (RWO) dataset which contains various real-word obstructions close to the camera. The experiments with our RWO dataset shows that our approach significantly improves the result compared to other methods.

Our main contributions are:

(1) We propose a novel repair feature module which contains **RSM** and **COA**. RSM first marks possible blur regions in features, then COA fixes the region by aggregating appearance information of objects.

(2) We propose a new data augmentation strategy called **DMD** which generates real semantic obstructions based on a dynamic binary mask. By applying DMD during training, models improve robustness to videos with large obstruction.

(3) By integrating our modules with previous methods, we demonstrate that our method achieves significant performance improvement on our RWO dataset and attains state-of-the-art results on multiple large occlusion benchmarks. Our method is the first to handle large obstructions in video frame interpolation.

## 2 RELATED WORK

Existing VFI methods can be classified into three categories: kernel-based, phase-based and flow-based approaches.

Kernel-based methods regard motion estimation as being joint with motion compensation. AdaConv [26] proposes estimating a pair of spatially-adaptive convolution kernels using a CNN and predicting intermediate flow by convolving input frames with the proposed kernels. A follow-up work [27] uses adaptive separable convolutions to reduce the large computational memory demand. Other alternatives improve it by estimating extra kernel offset vectors [2, 16], integrating deformable convolution to increase the receptive field [2], and proposing a multi-scale tailored loss function [31]. However, they are generally computationally expensive and inefficient in dealing with occlusion.

Earlier phase-based approaches (e.g. [24]) use phase information to learn the motion relationship, and they interpolate phase across the levels of a multi-scale pyramid. This approach works well with small motions but fails to handle challenging interpolation. PhaseNet [23] proposes a neural network that directly estimates the phase decomposition of the intermediate frame.

Previous flow-based works improve performance of occlusion scenes by various techniques including by estimating more robust optical flow for warping pixels [8, 10, 30, 39] or by determining occlusions with a depth map [1]. Recently, flow-based approaches have made good progress. CIAN [4] proposes to use channel attention and PixelShuffle to replace the extra flow estimation network. RIFE [8] proposes a sub-network IFNet that support arbitrary-timestep frame interpolation with the temporal input. FILM [33] proposes a multi-scale feature extractor that shares weights at all scales for large motion videos. IFRNet [15] proposes a lightweight encoder-decoder based network architecture for real-time applications. VFIformer [22] is a transformer-based U-net framework which enlarges the receptive field for capturing long-range information. Yoo et al. [40] proposed to use video object segmentation as an auxiliary task to train VFI models, which can learn extra segmentation information to interpolate frames with more precise object boundaries. Kim et al. [14] proposed to use event voxel grids as an additional input modality to capture the changes in pixel intensity asynchronously and with high temporal resolution, which can handle large and fast motions. Lee et al. [17] proposed a Figure-Text Mixing data augmentation technique for handling discontinuities in frames, which inserts static shapes and text. Unlike their approach, our data augmentation adds real-world scenes as obstructions and makes them move between frames. Plack et al. [32] propose enhancing frame rendering by estimating an error map and using it to

refine the frame quality through a second forward pass. EMA [41] proposed a transformer block that extracts motion and appearance information separately to better capture global motions compared with mixed feature map. BiFormer [28] proposed a transformer based architecture including blockwise bilateral cost volumes, local motion module and global motion refinement that handles 4K video frame interpolation. However, those methods still face the challenge of dealing with videos with large obstructions. In this paper, we mainly focus on handling close camera obstruction for real world frame interpolation.

## 3 METHOD

### 3.1 Overview

VFI is a technique to generate intermediate frames between two consecutive video frames $I_0, I_1 \in \mathbb{R}^{H \times W \times 3}$. Where $H$ and $W$ are the height and width, respectively. The general framework of flow-based VFI usually contains two steps. First, the network calculates intermediate optical flow $O_{t \to 0}$ and $O_{t \to 1}$ (t=0.5 in practice). Second, it predicts a residual $\delta I_t$ and a Mask $\mathcal{H}$ based on intermediate optical flow and input frames. The intermediate frame $I_t$ is estimated as:

$$\tilde{I}_0, \tilde{I}_1 = warp(I_0, O_{t \to 0}), warp(I_1, O_{t \to 1}) \tag{1}$$

$$I_t = \tilde{I}_0 * \mathcal{H} + \tilde{I}_1 * (1 - \mathcal{H}) + \delta I_t. \tag{2}$$

Most of the existing VFI networks [12, 15, 30, 41] are designed to deal with obstruction-free scenes. Our main goal is to develop an efficient and robust method for both obstruction-free scenes and scenes with obstructions. To achieve this, we propose an Obstruction Robustness Framework (ORF). The main idea is that we first identify ambiguous regions, then we explicitly recover them by looking at the structure of the original object in input images. In Section 3.2, we discuss how to identify ambiguous regions by the Region Similarity Map. Next, in Section 3.3, we propose a COA module, which captures accurate structure information from input frames and fills in the error regions. Finally, we propose a Dynamic Mask Distractions data augmentation method to generate obstructions in the training data

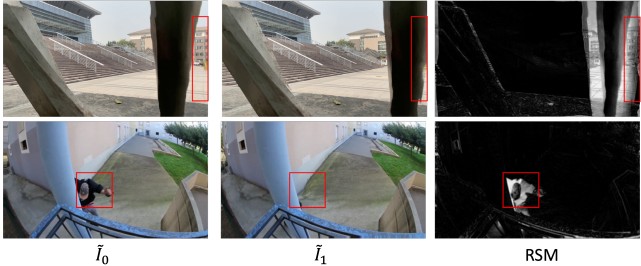

**Figure 3: Two examples of occlusions in a scene with obstructions. The first and second columns show the warped input frames. The pixels in the red rectangle fail to warp to the same position. Our Region Similarity Map clearly indicates the error.**

$\tilde{I}_0$        $\tilde{I}_1$        RSM

### 3.2 Region Similarity Map

Figure 3 shows two examples of warped frames with obstructions. The first and second columns show the warped input frames $\tilde{I}_0$ and $\tilde{I}_1$, respectively. The red rectangles highlight the regions where $\tilde{I}_0$ and $\tilde{I}_1$ should be identical or similar. However, we observed that those regions are not correctly aligned by optical flow which could cause large artifacts and distortions in the intermediate frame. The third column shows our proposed Region Similarity Map (RSM), which successfully measures the error regions between the warped frames. We define RSM $C$ as a 1-channel map, where the value of each pixel is in $(0, 1)$. Higher value indicates greater similarity between the corresponding pixels, and vice versa. In practice, we want to keep unambiguous pixels in warped frames, and only repair ambiguous pixels in $1 - C$.

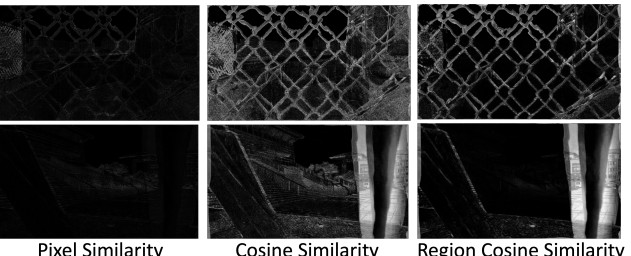

**Figure 4: A comparison of different similarity methods for scenes with obstructions. The region cosine similarity method is most stable.**

Pixel Similarity     Cosine Similarity     Region Cosine Similarity

To find the best measurement of RSM, we conducted experiments with three different methods: pixel similarity, cosine similarity and region cosine similarity. We extract feature maps $F_0 \in \mathbb{R}^{W \times H \times c}$ and $F_1 \in \mathbb{R}^{W \times H \times c}$ from $I_0$ and $I_1$, and warp $F_0$ and $F_1$ to $\tilde{F}_0$ and $\tilde{F}_1$ by using opitcal flow $O_{t \to 0}$ and $O_{t \to 1}$. We use $\tilde{F}_0$ and $\tilde{F}_1$ to calculate the similarity map as shown in Figure 4. The first column indicates the pixel similarity, which is computed by simply taking the pixel-wise difference between $\tilde{F}_0$ and $\tilde{F}_1$. Then averaging along the channel dimension to obtain a 1-channel map. It fails to highlight the most dissimilar regions. The second column is the cosine similarity. We treat the channels of each pixel as a vector with shape $1 \times c$ and calculate the cosine similarity between each pair of corresponding vectors. Cosine similarity emphasize the regions with large differences, but also introduces small scale noise which may mislead the network to focus on unimportant areas. Finally, we propose region cosine similarity. For each pixel in $\tilde{F}_0$, we calculate the similarity with its $n$ by $n$ neighbours and the corresponding region in $\tilde{F}_1$. The map only focuses on regions with large differences and avoids to highlight pixels with small motion. More visualizations of RSM can be found in the supplementary material.

### 3.3 Feature Repair

A naive way to reduce the artifacts and distortions in dissimilar regions is to apply a lightweight residual CNN-based module and hope that those errors at the feature level can be repaired by local information. Let $F_t$ be the intermediate feature with artifacts. The

recovered $\hat{F}_t$ can be written as.

$$\hat{F}_t = F_t * C + conv(F_t) * (1 - C) \qquad (3)$$

However, this method may not work well when dealing with large obstructions, as local features may not provide sufficient information. Therefore, we propose to use a transformer module that can model long-range dependencies and learn global context.

**Figure 5: The details of our COA: Each patch and its 8-neighborhood in the structure feature map $F_s$ are projected into one token. Attention is computed with the key $K$ from the inter-frame feature $F_t$. The correspondence will match if the patch of $F_t$ is similar to the patch of $f$ and its neighbors. The hidden features are calculated by the dot product of attention and values $V$, and repairs the inter-frame features in the explicit region indicated by the Region Similarity Map $C$.**

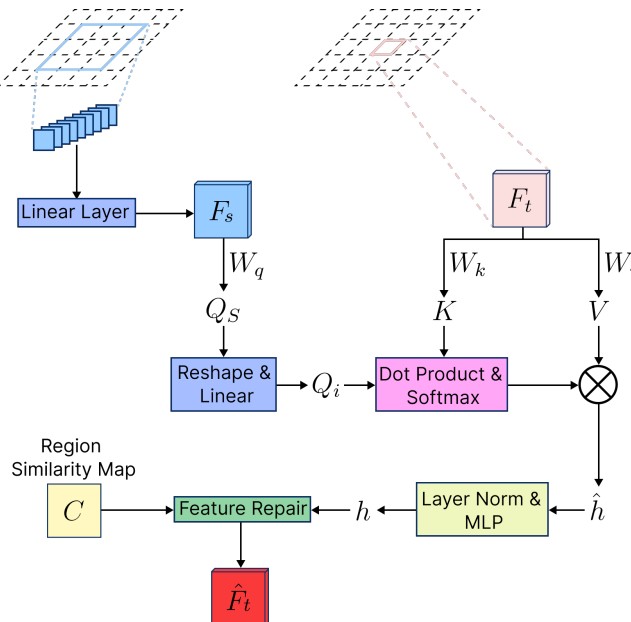

**Cross Overlap Attention.** As humans, we can easily recover a blurred object with a clear reference image that contains the same non-blurred object. We hypothesize that by comparing non-blurred objects from the original input features and blurred objects in synthetic features, the network can extract clearer structure information for blurred pixels. Our core idea is to allow the network to globally match the same objects, and aggregate the accurate structure information from input frame to synthetic features. Based on this idea, we adopt a transformer block [19] that can model long-range dependencies. Different from Inter-Frame Attention [41], which employs attention between two input frames and aims to enhance the extraction of appearance features. Our method applies cross-attention between input frame and synthetic frame to indicate accurate structure information only for the repair of blurred pixels.

The details are presented in Figure 5. We first extract structural information $F_s$ by concatenating $F_0$ and $F_1$ with lightweight CNN layers as most objects in an inter-frame can be found in neighboring

**Figure 6: The architecture of our model, which uses any optical flow based network as the VFI model. We take the input feature map from the encoder and use the feature repair module to fix the feature in the decoder.**

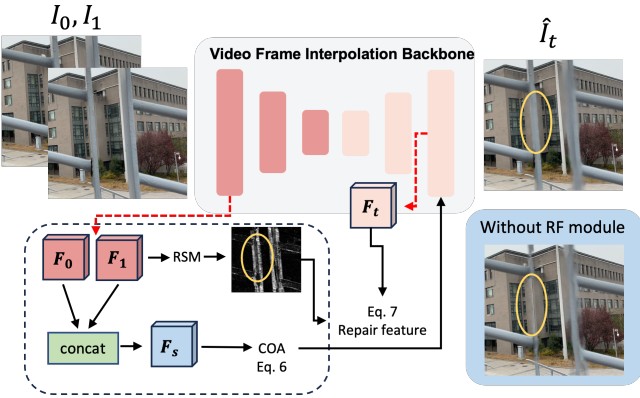

frames. We use $F_s$ to extract query vectors $Q_s \in \mathbb{R}^{N_q \times d}$, and inter-frame features $F_t$ to extract key vectors $K \in \mathbb{R}^{N_k \times d}$ and value vectors $V \in \mathbb{R}^{N_v \times d}$, where d is the feature dimension.

$$Q_s = F_s W_q, K = F_t W_k, V = F_t W_v \qquad (4)$$

Where $W_q$, $W_k$ and $W_v$ are projection matrices.

The attention matrix computed from the $Q_s$ and $K$ is most likely to match the same object. However, the patch to patch matching could be inefficient due to the blurring or artifacts in the synthetic feature, which could mislead the matching. To reduce the mismatch of the attention map, we project each patch $q$ and its 8-connected neighbors in Query $Q_s$ into one token, and then compute the attention with the corresponding patch $k$ in Key $K$. This implies that if patch $q$ and its neighbors are similar to $k$, they are more likely to belong to the same object. Specifically, for each patch $q$ in $Q_s$, we aggregate its 8-connected neighbors patches into a tensor $\hat{Q}_s \in \mathbb{R}^{N_q \times d \times 9}$. We then treated each patch $q_s \in \mathbb{R}^{d \times 9}$ in $\hat{Q}_s$ as a vector, and project it back to $\mathbb{R}^d$ by a linear projection to obtain $\hat{Q}_i \in \mathbb{R}^{N_q \times d}$. Finally, we compute the similarity between $\hat{Q}_i$ and $K$. The output $\tilde{h}$ can be formulated as

$$\tilde{h} = \text{Attention}(\hat{Q}_i, K, V) = \text{Softmax}\left(\frac{\hat{Q}_i K^T}{\sqrt{d}} + B\right) \cdot V \qquad (5)$$

where $B$ is the position bias [6]. We incorporate cross overlap attention into the transformer block. Following [19], we also apply normalization (LN) and a multi-layer perceptron (MLP) after attention as

$$h = COA(F_t, F_s) = MLP(LN(\tilde{h})). \qquad (6)$$

**Feature repair with RSM** Figure 6 illustrates our proposed method for repairing the intermediate feature with RSM and the COA module. Given a baseline VFI network, it extracts the features $F_0$ and $F_1$ from the encoder. We calculate RSM in Section 3.2. Then, we apply the COA module to refine the intermediate feature by using Equation 6. We use the COA module to refine the features in

dissimilar regions $(1 - C)$. The refined features $\hat{F}_t$ are calculated as.

$$\hat{F}_t = F_t * C + h * (1 - C) \tag{7}$$

We choose the optimal intermediate feature layer for each network based on the performance of the RSM and the COA module. The details of our layer selection are provided in the supplementary material.

**Figure 7: Schematic View of our Dynamic Mask Distractions (DMD) Data Augmentation.**

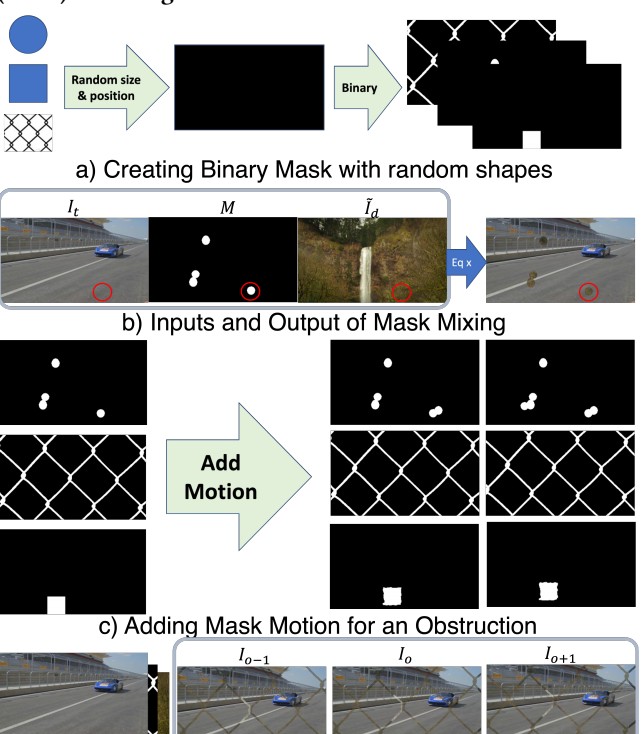

a) Creating Binary Mask with random shapes

b) Inputs and Output of Mask Mixing

c) Adding Mask Motion for an Obstruction

d) Frame Triplet for Supervised VFI

## 3.4 Dynamic Mask Distractions

Most of previous studies applied low-level modifications such as flip augmentation along spatial and temporal axes. Lee et al. [17] propose a Figure-Text Mixing data augmentation technique that artificially inserts static shapes and texts. However, those augmentations are not sufficient to handle videos with dynamic obstructions. We propose a new data augmentation method for frame interpolation, called Dynamic Mask Distractions (DMD). The main idea of DMD is to generate obstructions on original video with real scenes. It has been shown that synthesizing obstructions such as reflections and fences can make the network adapt to real-world scenes with obstructions [18, 34]. However, those methods need to use extra dataset (e.g., fence data) and only fit specific obstructions. In real-world settings, models frequently encounter unforeseen obstructions. Rather than training the model to identify specific obstructions, our objective is to enable the model to recognize general

patterns of obstruction. Thus, we generate images with obstructions by using real distractions.

**Create Binary Mask.** We first collect random shapes such as circles, rectangles, fences, etc. Then, we generate a binary mask $M$ based on the random positions of these shapes, as shown in Figure 7 (a). Given an image $I_t$, an image $\tilde{I}_d$ from a different scene and a randomly selected binary mask $M$. The new obstructed image $I_b$ can be presented as

$$I_b = I_t * M * \lambda + \tilde{I}_d * M * (1 - \lambda) + I_t * (1 - M), \tag{8}$$

where $\lambda \in (0, 1)$ is a random ratio.

Previous methods [18, 34] synthesize obstructions in a single image. In a video, obstructions usually move. The motion may be continuous or discontinuous depending on the physical properties of the obstruction and the imaging setup. Thus, we propose a simple but effective way to simulate the motion of obstructions.

**Add Motion.** We simulate three categories of motions for obstructions: translation, deformation, and discontinuous motion. Translation refers to the movement of rigid obstructions in a random direction, e.g., motion of fences close to the camera. Deformation occurs when non-rigid obstructions change their shape, e.g., dirt on a window or the lens itself. Discontinuous motion describes the sudden appearance or disappearance of obstructions, e.g., from raindrops on a lens, or due to specular reflection.

As shown in Figure 7 c), given the binary mask in Equation 8, we generate masks for the other frames by applying different motions to the first mask. For translation, we randomly add an offset to the position of the obstruction and move the mask for the second and third images. For deformation, we use elastic deformation [36], which modifies the shape of the obstruction by applying random displacements to a grid of pixels. The mask is then interpolated to fit the deformed grid, creating a distorted version of the obstruction. For discontinuous motion, we generate a random number of obstructions of a certain shape, and then randomly add or delete obstructions in the mask for the second and third image consecutively.

In practice, let $(I_{t-1}, I_t, I_{t+1})$ be a triple of subsequent images from a video. Then the obstructed version of this triple is $(I_{b-1}, I_b, I_{b+1})$. Figure 7 d) is an example of training data.

## 4 EXPERIMENTS

### 4.1 Datasets and Implementation Details

**Training and Evaluation.** Our model is trained on the Vimeo90K [39] training set. For hyperparameters, such as learning rates, batch sizes, and optimizers, we follow the settings reported in the original papers of each baseline network. For the data augmentation, we apply random cropping and flipping to the input frames along the horizontal, vertical, and temporal dimensions. We also incorporate the proposed data augmentation DMD, which has been described in Section 3.2.

We evaluate our method on four recent state-of-the-art (SOTA) flow-based VFI methods: RIFE [8], IFRNet [15], VFIFormer [22], and EMA [41]. To demonstrate the effectiveness of each component of our method, We first conducted an ablation study on each DMD, RSM and COA. Then, we compare our method with the previous SOTA methods on following three benchmark datasets:

**Figure 8: Qualitative results on SNU test dataset.**

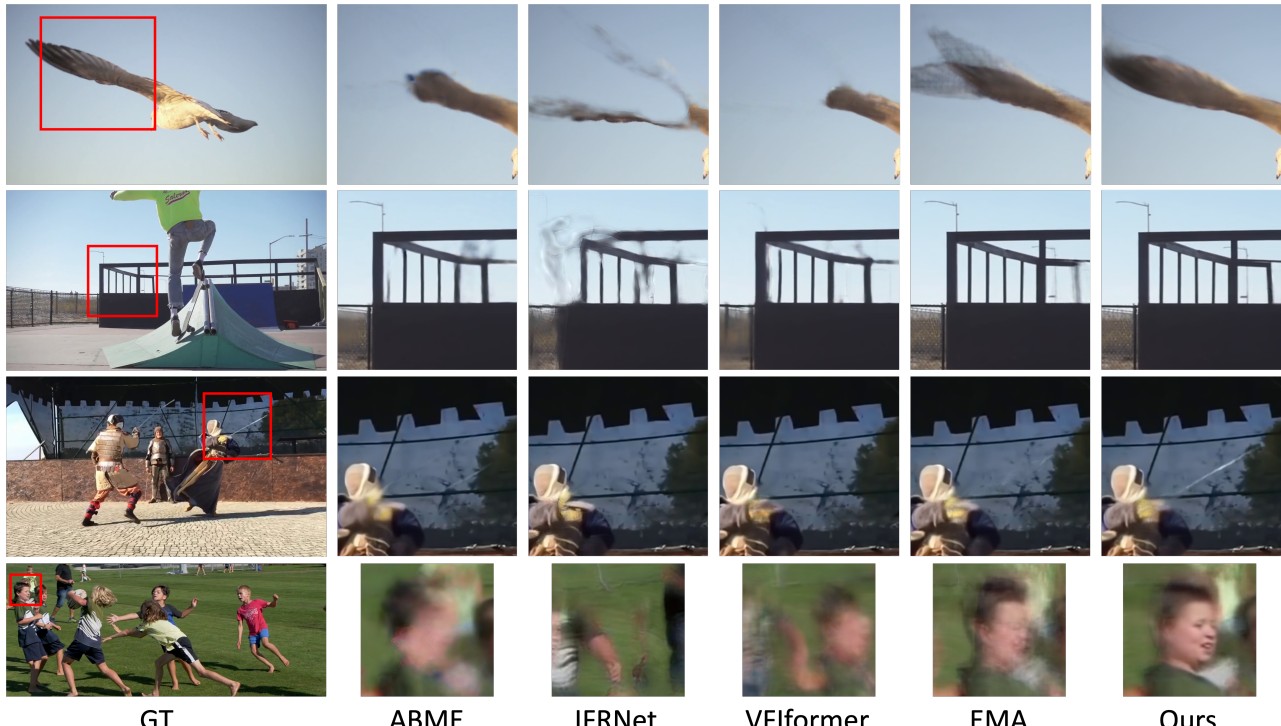

| GT | ABME | IFRNet | VFIformer | EMA | Ours |

**Vimeo90K.** It contains 3782 triplets for testing.

**SUN-Film.** It contains 1240 triplets for testing. It is spilt into four different parts, Easy, Medium, Hard and Extreme. The latter two categories are characterized by large motion and increased occlusions.

**RWO.** It is a real-world obstruction dataset, which is a new dataset that we construct from real-world obstructions. It contains 61 triples for testing. More detail can be found in supplementary material. We measure the performance of each algorithm by comparing Peak Signal-to-Noise Ratio (PSNR) and Structural Similarity (SSIM) [37].

## 4.2 Comparison with the State-of-the-Art

We compare our methods with previous state-of-the-art VFI methods, including AdaCof [16], CAIN [4], RIFE [8], ABME [30], IFR-Net [15], VFIformer [22] and EMA [41]. We add our methods to the two highest performing previous methods: VFIformer-ORF and EMA-ORF. As shown in Table 1, our methods outperform the previous algorithms on the RWO dataset by a considerable margin. Although, VFIformer has a lower performance than ABME and EMA on the RWO dataset, our VFIformer-ORF improves the results and outperforms all previous methods. The results clearly demonstrate the limitations of previous methods and the robustness of our approach. Moreover, our EMA-ORF also improves large occlusion cases in SNU-FLIM hard and extreme. It even shows competitive performance with SOTA methods on other benchmarks that do not

contain large obstructions. Thus, our method could be a potential solution to deal with large obstruction in the real-world.

## 4.3 Ablation Study

To evaluate the effectiveness of our DMD and RSM to deal with large obstruction and occlusion data, we conduct the ablation study in Table 2. We make the following observations: 1. DMD data augmentation slightly enhances the accuracy on SNU-FILM Hard and Extreme datasets which contain large motion and large obstructions. Moreover, it clearly improves the performance in the presence of considerable obstructions. 2. The RSM module further improves both, large motion and large obstruction data. We attribute this to the fact that SNU-FILM hard and extreme have many large displacements that are difficult to warp to the same position. The RSM can efficiently capture those parts (see Figure 8). 3. Both components have demonstrated efficiency across all four methods.

**Ablation for Region Similarity Map.** Here, we discuss the detail of RSM. We first compare the different measurements and their corresponding results. As shown in Figure 10, we found pixel similarity failed to highlight the error region, cosine similarity can find the error region but with some noise which degrades the output image. Region cosine similarity reduces noise and produces a clearer boundary in the error region, resulting in the most realistic output image. Table 3 shows the results of our EMA-ORF with different similarity measurement where region cosine similarity achieves the best results.

Figure 9: Qualitative results on our RWO dataset

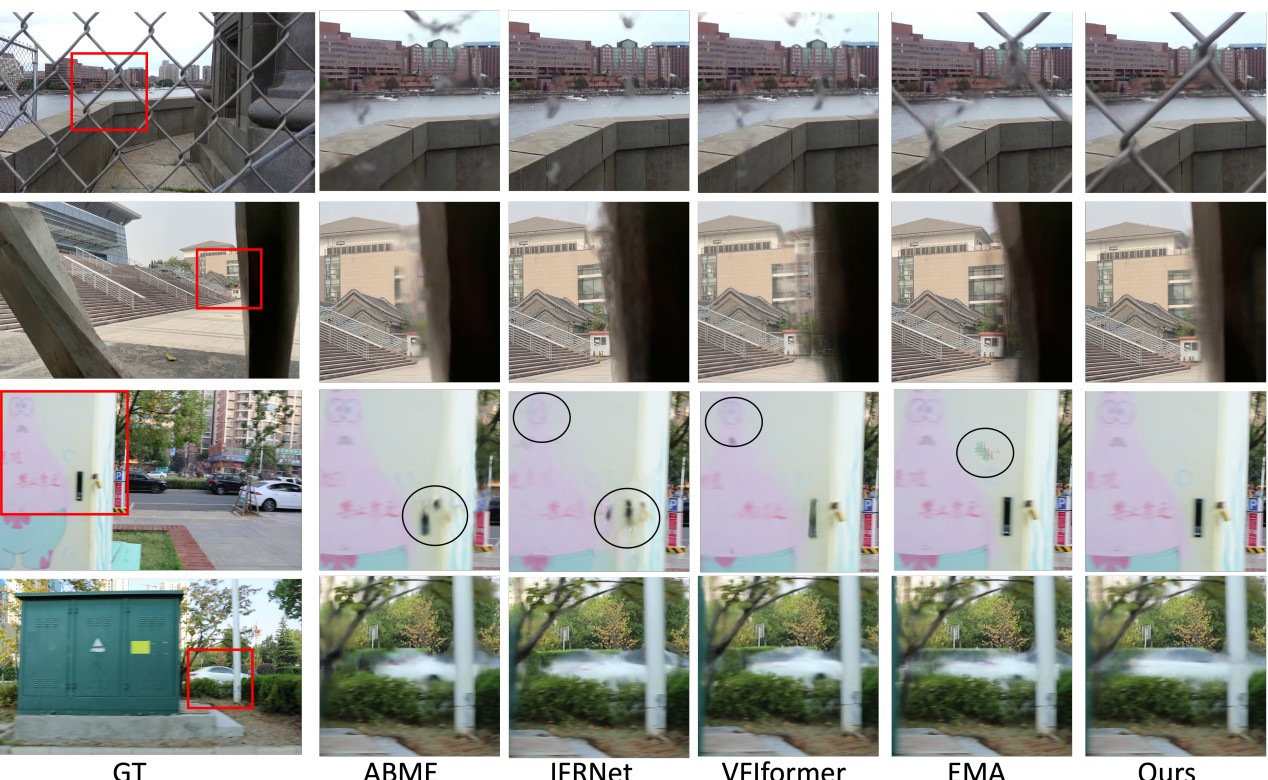

GT  ABME  IFRNet  VFIformer  EMA  Ours

Table 1: Quantitative comparison of VFI methods on three datasets. The best result is in red, and the second best is in blue. "-ORF" indicates our Obstruction Robustness Framework.

| Method | Vimeo90K | | SNU-FILM EASY | | SNU-FILM MED | | SNU-FILM Hard | | SNU-FILM EX | | RWO | |
|---|---|---|---|---|---|---|---|---|---|---|---|---|
| | PSNR | SSIM | PSNR | SSIM | PSNR | SSIM | PSNR | SSIM | PSNR | SSIM | PSNR | SSIM |
| AdaCof[16] | 34.47 | 0.9730 | 39.80 | 0.9900 | 35.05 | 0.9754 | 29.46 | 0.9244 | 24.31 | 0.8439 | 26.43 | 0.8608 |
| CAIN[4] | 34.65 | 0.9729 | 39.89 | 0.9900 | 35.61 | 0.9776 | 29.90 | 0.9292 | 24.78 | 0.8507 | 27.19 | 0.8805 |
| RIFE[8] | 35.61 | 0.9779 | 39.99 | 0.9904 | 35.68 | 0.9786 | 30.08 | 0.9327 | 24.83 | 0.8533 | 27.52 | 0.8837 |
| ABME[30] | 36.18 | 0.9805 | 39.59 | 0.9901 | 35.77 | 0.9789 | 30.58 | 0.9364 | 25.42 | 0.8639 | 28.93 | 0.9029 |
| IFRNet[15] | 35.80 | 0.9794 | 40.03 | 0.9905 | 35.94 | 0.9793 | 30.40 | 0.9358 | 25.05 | 0.8587 | 28.25 | 0.8939 |
| VFIformer[22] | 36.50 | 0.9816 | 40.13 | 0.9907 | 36.09 | 0.9799 | 30.67 | 0.9378 | 25.43 | 0.8643 | 28.77 | 0.8994 |
| EMA[41] | 36.64 | 0.9819 | 39.77 | 0.9908 | 35.98 | 0.9801 | 30.93 | 0.9395 | 25.69 | 0.8663 | 28.97 | 0.8978 |
| VFIformer-ORF | 36.50 | 0.9814 | 40.11 | 0.9905 | 36.07 | 0.9794 | 30.87 | 0.9396 | 25.73 | 0.8694 | 29.20 | 0.9043 |
| EMA-ORF | 36.69 | 0.9821 | 39.96 | 0.9909 | 36.14 | 0.9802 | 31.10 | 0.9419 | 25.95 | 0.8722 | 29.56 | 0.9081 |

**Ablation for Cross Overlap Attention.** According to Equation 7, we aim to obtain accurate geometric information to correct the error regions indicated by the RSM. In this context, we conducted experiments with different technologies: residual convolutional layers, self-attention, inter-frame attention [41] and our COA. In Figure 11, we find that the convolution layer fails to recover the region. This failure is attributed to the convolution layer capturing only local information, which is insufficient to recover large error regions. Subsequently, the self-attention mechanism recovered in

an incorrect region. The Inter-Frame Attention (IFA) recovers parts of error regions but still suffers from blurry boundaries because IFA does not bring the corresponding information between input and synthetic frames. Conversely, our Cross-Overlap Attention (COA) successfully recoveres it by building the correspondence directly between the input and synthetic images. In Table 4, our COA outperforms the other three methods across all three benchmarks, indicating its efficiency.

**Table 2: Ablation study for module components. Effect of applying DMD and RSM on the performance of the proposed model. "-D" indicates using DMD and "-DR" indicates using DMD and RSM. (Note: RSM is applied by Equation 3 in this Table. COA is evaluated in Table 1).**

| Method | SNU-FILM Hard | | SNU-FILM EX | | RWO | |
|---|---|---|---|---|---|---|
| | PSNR | SSIM | PSNR | SSIM | PSNR | SSIM |
| RIFE [8] | 30.08 | 0.9327 | 24.83 | 0.8533 | 27.52 | 0.8837 |
| RIFE-D | 30.14 | 0.9329 | 24.89 | 0.8548 | 27.73 | 0.8863 |
| RIFE-DR | **30.23** | **0.9341** | **25.01** | **0.8569** | **27.82** | **0.8874** |
| IFRNet [15] | 30.40 | 0.9358 | 25.05 | 0.8587 | 28.25 | 0.8939 |
| IFRNet-D | 30.49 | 0.9357 | 25.17 | 0.8587 | 28.40 | 0.8950 |
| IFRNet-DR | **30.54** | **0.9363** | **25.24** | **0.8592** | **28.53** | **0.8962** |
| VFIformer [22] | 30.67 | 0.9378 | 25.43 | 0.8643 | 28.77 | 0.8994 |
| VFIformer-D | 30.70 | 0.9380 | 25.47 | 0.8659 | 29.02 | 0.9014 |
| VFIformer-DR | **30.80** | **0.9383** | **25.64** | **0.8679** | **29.11** | **0.9028** |
| EMA [41] | 30.93 | 0.9395 | 25.69 | 0.8663 | 28.97 | 0.8978 |
| EMA-D | 30.95 | 0.9397 | 25.73 | 0.8679 | 29.26 | 0.9032 |
| EMA-DR | **31.02** | **0.9407** | **25.86** | **0.8701** | **29.33** | **0.9046** |

**Figure 10: Visualization of different RSM measurement.**

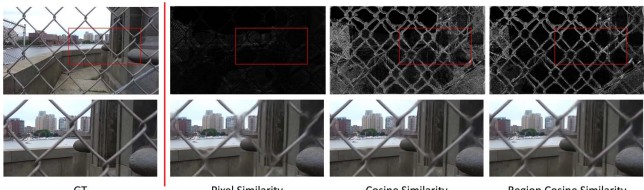

GT | Pixel Similarity | Cosine Similarity | Region Cosine Similarity

**Table 3: Quantitative results of different RSM measurement.**

| Method | SNU-FILM Hard | | SNU-FILM EX | | RWO | |
|---|---|---|---|---|---|---|
| | PSNR | SSIM | PSNR | SSIM | PSNR | SSIM |
| Pixel Similarity | 31.04 | 0.9409 | 25.76 | 0.8690 | 29.24 | 0.9037 |
| Cosine Similarity | 31.07 | 0.9418 | 25.90 | 0.8713 | 29.46 | 0.9072 |
| RCS | **31.10** | **0.9419** | **25.95** | **0.8722** | **29.56** | **0.9081** |

**Figure 11: Visualization of different repair module.**

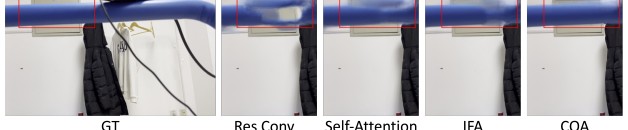

GT | Res Conv | Self-Attention | IFA | COA

### 4.4 Qualitative Evaluation

**Clean Scenes.** We select samples from the SNU-Film dataset to compare the quality of different VFI methods. We compare our method with ABME, IFRNet, VFIformer and EMA in Figure 8. The red rectangles indicate challenging regions where unlike previous methods our method achieve accurate results. Specifically, the first

**Table 4: Quantitative results of different repair module.**

| Method | SNU-FILM Hard | | SNU-FILM EX | | RWO | |
|---|---|---|---|---|---|---|
| | PSNR | SSIM | PSNR | SSIM | PSNR | SSIM |
| Res Conv | 31.02 | 0.9407 | 25.86 | 0.8701 | 29.33 | 0.9046 |
| Self-Attention | 31.03 | 0.9411 | 25.88 | 0.8709 | 29.41 | 0.9053 |
| IFA[41] | 31.08 | 0.9416 | 25.92 | 0.8717 | 29.52 | 0.9065 |
| COA | **31.10** | **0.9419** | **25.95** | **0.8722** | **29.56** | **0.9081** |

row shows large motion where IFRNet and EMA warp pixels to different positions which causes ghosting. ABME and VFIformer fail to predict this region. Our method preserves the details and the motion consistency. The third row is a tiny object case, where all four previous methods fail to reconstruct the thin sword. However, our method successfully preserves it. The fourth row is an occlusion case, where the person's face is visible in one input frame but partially visible in the other. Other methods produce blurry or distorted faces due to mismatches while our method generates the clearest results. We suppose the reason is that RSM successfully identified the face, and COA effectively recovered it.

**Scenes with Obstructions.** We further visualize examples in our RWO dataset (Figure 9). We observe two issues of previous methods: 1) They sometimes fail to maintain the structure of the obstruction; and 2) They often have difficulty distinguishing the boundary of the obstruction and the background object and produce artifacts in the frames. In the first row, ABME, IFRNet and VFIformer fail to generate fences, and EMA partially fails, but our method succeeds in reconstructing it. In the second row, all four previous methods mixed the boundary between the pillar and the house. In comparison, our method shows a clearer shape. These examples demonstrate that our method successfully reduces errors in large obstruction cases.

## 5 LIMITATIONS

Our method can handle most videos with large obstructions. However, it still fails in some cases, such as videos at night. The main reason is that when pixels are all similar (darkness) in a scene, our method does not always capture ambiguous pixels/regions. Another case is the use of a flash in a video, which could lead to large dissimilarity for the whole frame and not a dissimilar region which breaks our hypothesis. The COA module would have difficulties to reconstruct the whole frame. However, our method is indeed robust to most cases of obstructions. We hope to address the remaining issues in future research.

## 6 CONCLUSION

In this paper, we propose a novel VFI method that can handle videos with both obstructions and clean frames. We introduce a novel feature repair module, which first detects dissimilar regions of the two warped input frames, then repairs those regions by cross overlap attention (COA). We also propose a new data augmentation method (DMD) to make VFI networks robust for obstructions without extra images. Extensive experiments demonstrate that our method shows remarkable improvement on obstruction datasets, and also performs well on standard benchmarks.

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
