# OpenReview forum: "Learning to Handle Large Obstructions in Video Frame Interpolation"
_acmmm.org/ACMMM/2024/Conference — MM2024 Poster_

### Official Review · Reviewer_jrpR · 2024-05-06

**Rating:** 4
**Confidence:** 2

**Summary:**

This paper addresses the challenge of video frame interpolation (VFI) in the presence of large obstructions, which often cause blur and artifacts, making the video discontinuous. To overcome this, the authors propose the Obstruction Robustness Framework (ORF), comprising a feature repair module and a data augmentation strategy. The feature repair module identifies and repairs ambiguous pixels using a region similarity map and cross-overlap attention module, while the data augmentation strategy enables dynamic obstruction handling without additional data. This approach, applied to existing VFI networks, not only improves results on standard benchmarks but also significantly enhances interpolation quality for obstructed videos, marking a contribution in the field.

**Strengths:**

1. The method is clearly written, the solution is elegant and the result seems well.

**Limitations:**

1. I am not sure whether the code and model will be released. Consequently, I am concerned about the reproducibility and potential impact of this work.

2. The description from line 321 to line 341 for Figure 4 is unclear. Incorporating formulas to illustrate the process would enhance comprehension and clarity.

3. Some evaluation results in Table 1 do not rank as the best compared to other methods. The authors may need to analyze these discrepancies.

**Suitability:**

2

---

### Official Review · Reviewer_SGba · 2024-05-24

**Rating:** 3
**Confidence:** 4

**Summary:**

This paper proposes an Obstruction Robustness Framework (ORF) that can enhance the robustness of existing video frame interpolation networks in the face of large obstructions. The main contributions of the proposed ORF are a feature repair module and a data augmentation strategy. By combining the ORF with previous state-of-the-art backbones, the proposed method not only improves the results in original benchmarks but also significantly enhances the interpolation quality for videos with obstructions.

**Strengths:**

1. This paper is well-written and easy to understand.
2. This paper focuses on addressing the problem of video frame interpolation in occluded scenarios and proposes a refinement module that can be integrated with existing methods. By incorporating this refinement module, existing video frame interpolation methods can achieve better results in the presence of occlusions. This paper provides new insights and values for the video frame interpolation task.

**Limitations:**

1. Although this paper specifically addresses video frame interpolation in occluded scenarios, I observed that the interpolation results in the grid portions of the demo videos provided by the authors are still quite jittery, and the overall performance is not very satisfactory.
2. The proposed module can be integrated into some previous state-of-the-art models, which inevitably increases the original models' parameters and runtime to some extent. Would simply increasing the model parameters also achieve better results? The authors should provide a comparison of the parameter count and runtime for different models.
3. When integrating the proposed refinement module into state-of-the-art models, are all the parameters trained during the process, or is it sufficient to fine-tune only the proposed refinement module? If training all the parameters is necessary, the proposed refinement module might not be as convenient to use.
4. The proposed refinement module first computes a Region Similarity Map (RSM) and then uses Cross Overlap Attention (COA) to fill in the missing information.  Why in Table 2, COA is not included?
5. Although the paper claims that the proposed method can improve frame interpolation results, Table 1 shows that the increase in PSNR is not very significant, with some improvements being as small as 0.0x.
6. This paper primarily discusses how to address occlusion problems in video frame interpolation methods based on backward warping. However, besides backward warping, there is also forward warping, as discussed in several other papers[1][2]. Forward warping can introduce hole-filling issues, which theoretically could also be addressed by the proposed refinement module. The authors should discuss methods related to forward warping in the paper.

[1]. Li, Yu, et al. "Hybrid warping fusion for video frame interpolation." International Journal of Computer Vision 130.12 (2022): 2980-2993.
[2]. Niklaus, Simon, and Feng Liu. "Softmax splatting for video frame interpolation." Proceedings of the IEEE/CVF conference on computer vision and pattern recognition. 2020.

**Suitability:**

3

---

### Official Review · Reviewer_fXRQ · 2024-05-24

**Rating:** 4
**Confidence:** 4

**Summary:**

This paper focuses on video frame interpolation for videos that contain large obstructions. The proposed Obstruction Robustness Framework (ORF) is devised to improve the robustness of video frame interpolation networks by introducing a feature repair module and a data augmentation strategy into backbones built upon existing methods. However additional clarification and comparisons may be needed for better understanding and evaluation.

**Strengths:**

1. A new dataset is proposed to test the performance of the proposed method.
2. The improvement against existing methods being used as backbones is achieved with only a slight increase of parameters.

**Limitations:**

1. The challenges of video frame interpolation due to occlusions are tackled in existing works. “occlusion” covers “obstruction” that refers to a type of blockages, it may be necessary to notice that the challenge of “obstruction” being addressed in this paper is in fact the same as the challenge of “occlusion” being addressed in previous works.
2. Since the proposed method utilizes existing methods as backbones, comparisons with these methods on more benchmarks, i.e., Middlebury and Xiph that have being used by them may better demonstrate the improvements.
3. The qualitative result of the proposed method shown on the last row in Figure 9 still contains a blurry car. The improvements are limited compared with existing method.

**Suitability:**

2

---

### Meta-Review · Area_Chair_yc3f · 2024-06-27

**Recommendation:** Accept (Poster)
**Confidence:** 5

**Metareview:**

Reviewers acknowledged the good results and new dataset of the paper. The rebuttal was successful, with two reviewers raising their scores, and now the paper received all accept recommendations. The area chairs agree with this recommendation and are pleased to inform you that your paper has been accepted. The authors are required to incorporate the reviewers' suggestions into the camera-ready version. Specifically, there are remaining issues from Reviewer fXRQ to be addressed: (1) add #params and runtime comparison; (2) add necessary citations and discussions with reference [1][2].